**Data Availability Statement:** All relevant data are within the manuscript and its Supporting Information files.

# Feasibility of integrating canine olfaction with chemical and microbial profiling of urine to detect lethal prostate cancer

**Claire Guest**[1⊙]*, **Rob Harris**[1⊙], **Karen S. Sfanos**[2,3⊙], **Eva Shrestha**[2‡], **Alan W. Partin**[3‡], **Bruce Trock**[3⊙], **Leslie Mangold**[3‡], **Rebecca Bader**[4⊙], **Adam Kozak**[4‡], **Scott Mclean**[4‡], **Jonathan Simons**[5⊙], **Howard Soule**[5⊙], **Thomas Johnson**[5⊙], **Wen-Yee Lee**[6⊙], **Qin Gao**[6], **Sophie Aziz**[1], **Patritsia Maria Stathatou**[7‡], **Stephen Thaler**[8⊙], **Simmie Foster**[9], **Andreas Mershin**[7⊙]*

1 Medical Detection Dogs, Milton Keynes, United Kingdom, 2 Department of Pathology, Johns Hopkins University School of Medicine, Baltimore, Maryland, United States of America, 3 Department of Urology, James Buchanan Brady Urological Institute, Johns Hopkins University School of Medicine, Baltimore, Maryland, United States of America, 4 Cambridge Polymer Group, Cambridge, Massachusetts, United States of America, 5 Prostate Cancer Foundation, Santa Monica, California, United States of America, 6 Department of Chemistry and Biochemistry, University of Texas at El Paso, El Paso, Texas, United States of America, 7 The Center for Bits and Atoms, Massachusetts Institute of Technology, Cambridge, Massachusetts, United States of America, 8 Imagination Engines, St. Charles, Missouri, United States of America, 9 Department of Psychiatry, Harvard Medical School and Massachusetts General Hospital, Boston, Massachusetts, United States of America

⊙ These authors contributed equally to this work.
‡ These authors also contributed equally to this work.
* mershin@mit.edu (AM); claire.guest@medicaldetectiondogs.org.uk (CG)

## Abstract

Prostate cancer is the second leading cause of cancer death in men in the developed world. A more sensitive and specific detection strategy for lethal prostate cancer beyond serum prostate specific antigen (PSA) population screening is urgently needed. Diagnosis by canine olfaction, using dogs trained to detect cancer by smell, has been shown to be both specific and sensitive. While dogs themselves are impractical as scalable diagnostic sensors, machine olfaction for cancer detection is testable. However, studies bridging the divide between clinical diagnostic techniques, artificial intelligence, and molecular analysis remains difficult due to the significant divide between these disciplines. We tested the clinical feasibility of a cross-disciplinary, integrative approach to early prostate cancer biosensing in urine using trained canine olfaction, volatile organic compound (VOC) analysis by gas chromatography-mass spectroscopy (GC-MS) artificial neural network (ANN)-assisted examination, and microbial profiling in a double-blinded pilot study. Two dogs were trained to detect Gleason 9 prostate cancer in urine collected from biopsy-confirmed patients. Biopsy-negative controls were used to assess canine specificity as prostate cancer biodetectors. Urine samples were simultaneously analyzed for their VOC content in headspace via GC-MS and urinary microbiota content via 16S rDNA Illumina sequencing. In addition, the dogs' diagnoses were used to train an ANN to detect significant peaks in the GC-MS data. The canine olfaction system was 71% sensitive and between 70–76% specific at detecting Gleason 9 prostate cancer. We have also confirmed VOC differences by GC-MS

**Funding:** This work was funded by the Prostate Cancer Foundation Grant (18PILO02) received by CG, AM, and KS. PCF provided partial salary support for authors CG, RH, PMS, AM, and had a role in the study design and preparation of the manuscript, but had no role in the data collection and analysis or the decision to publish. The National Cancer Institute of the National Institutes of Health provided support for WYL and QG (Award Number SC1CA245675). Imagination Engines, Inc. provided support in the form of salary for ST. The NCI NIH and Imagination Engines, Inc. played a role in study design, analysis, decision to publish, and preparation of the manuscript. The specific roles of these authors are articulated in the 'author contributions' section.

**Competing interests:** The authors have read the journal's policy and have the following competing interests: ST is a paid employee of Imagination Engines, Inc. This does not alter our adherence to PLOS ONE policies on sharing data and materials. There are no patents, products in development or marketed products associated with this research to declare.

and microbiota differences by 16S rDNA sequencing between cancer positive and biopsy-negative controls. Furthermore, the trained ANN identified regions of interest in the GC-MS data, informed by the canine diagnoses. Methodology and feasibility are established to inform larger-scale studies using canine olfaction, urinary VOCs, and urinary microbiota profiling to develop machine olfaction diagnostic tools. Scalable multi-disciplinary tools may then be compared to PSA screening for earlier, non-invasive, more specific and sensitive detection of clinically aggressive prostate cancers in urine samples.

## Introduction

Prostate cancer is the leading type of non-skin cancer in the US, is the second most prevalent cancer worldwide, and has overtaken breast cancer in total deaths caused in the UK. Approximately 1 in 9 men will be diagnosed with prostate cancer at some point in their lives. Early biomarker detection of prostate cancer has been controversial, as the widely used Prostate Specific Antigen (PSA) screening test may miss clinically significant cancer in men with normal PSA levels, may over-diagnose men with clinically insignificant cancer, and erroneously detects benign conditions such as benign prostatic hyperplasia (BPH) and prostatitis [1]. There is an urgent need for non-invasive, more sensitive and specific diagnostic technologies. In particular, what is needed is a prostate cancer diagnostic tool to allow differentiation of potentially lethal, high Gleason Grade cancers with metastatic potential from indolent, low-grade cancers that patients would die with and not from.

One potential method towards improved prostate cancer diagnosis that has been receiving increased attention is trained canine olfaction. Over the past three decades, trained dogs have been shown to be capable of detecting various human diseases including many types of cancer by scent [2]. There are also several published case reports of untrained dogs spontaneously showing interest in skin cancer on their owners. In 1989, Williams and Pembroke wrote of a patient whose dog persistently sniffed a mole on her leg. The dog's excessive interest in the mole prompted the patient to visit a clinician, who identified the mole as a malignant melanoma [3]. In 2001, Church and Williams reported a man whose dog constantly sniffed at a patch of eczema on his leg, which after excision was found to be a basal cell carcinoma [4]. In 2013, Campbell et al. described a case in which a man's dog persistently licked a lesion behind his right ear, which was later confirmed to be malignant melanoma [5]. In each of these cases, the dog was apparently able to detect a signal of interest in the smell emanating from the skin close to the affected area. This supposition was supported by a dog trained to detect melanoma [6]. After a proof-of-principle study in bladder cancer was published in 2004 [7], an increasing number of studies investigating the ability of trained dogs to accurately detect cancer has appeared in the literature. These have included studies in the detection of lung, breast, ovarian, bladder, and prostate cancer [8–17]. In 2015, Taverna et al. published a pivotal paper on the canine detection of prostate cancer from urine [18]. This study included 362 cases and 540 healthy controls, with a striking mean sensitivity (2 dogs) of 99% and mean specificity (2 dogs) of 98%. These findings support the premise that olfactory detection of prostate cancer holds the promise of rapid and non-invasive diagnosis.

Given the limited availability of trained canines, in the present study we begin to explore what the dogs may be detecting and whether an artificial neural network (ANN) potentially deployed in conjunction with machine olfaction might be the tools to replicate the dogs' early detection capability. Previous studies have asked whether the odor of cancer in urine is

represented by one or a set of specific volatile organic compounds (VOCs), many of which elicit an olfactory response (i.e. they are "odorants"). Likewise, the urinary "volatilome" (the compilation of volatile metabolites as well as other volatile organic and inorganic compounds present), has also shown promise in prostate cancer diagnosis. In a recent paper by Lima et al., urinary VOC analysis distinguished prostate cancer cases with 78% sensitivity, and 94% specificity [19]. The urinary microbiome also contributes to VOC production [20], so it is reasonable to ask whether an integrative approach yields advantages. However, VOC analysis via gas or liquid chromatography (GC or LC) coupled to mass spectrometry (MS) or other analytical methods that rely on identifying compounds still present significant limitations as the signature scent of cancer might depend on a combinatorial mixture in perceptual space [21] as opposed to any specific set of individual odorants (as is the case with some scents [22]) increasing difficulty of standardization, scale up and practical deployment. Additional problems with chemical analytics include difficulty of access to expensive equipment with performance variabilities.

The limitations of current diagnostic methods and of molecular VOC analysis by gas chromatography-mass spectroscopy (GC-MS) led us to question if integrating a multiparametric approach could 1) lead to better diagnosis, 2) lead to a better understanding of the underlying disease pathology, and 3) illuminate the way towards machine olfaction-based urinary screening and diagnosis that are also receiving increasing attention [23]. Rather than rely solely on one method (canine detection, volatilomics, urinary microbiota profiling) to improve diagnostic efficacy, in this pilot study, we sought to combine the strengths of each to create new insight into how further integrative diagnostic developments can be made. To do this, we simultaneously submitted urine samples of patients with or without biopsy-proven prostate cancer for detection by trained canine olfaction, VOC identification by GC-MS, and microbiota profiling. We further trained an ANN on GC-MS data using the canine diagnoses. To our knowledge, this is the first study to profile both urinary VOCs and urinary microbial populations in the same urine samples. Our results show the feasibility and characterize the challenges to overcome in a cross-disciplinary approach to prostate cancer diagnosis using urinary VOCs and point towards development of practical machine olfaction-based diagnostics.

## Materials and methods

The experimental plan and study design are illustrated in Fig 1. A detailed description of the animal training facilities and canine training protocol are provided in the S1 Methods.

### Urine sample collection and patient characteristics

United States (US)-originated urine samples used in the final canine training and all GC-MS and canine testing, were obtained under a Johns Hopkins University (JHU) Medicine Institutional Review Board (IRB) approved protocol with written informed consent. For early training and canine testing the UK-originated samples were collected from participants identified by a member of the clinical team who provided them a verbal explanation and information sheet outlining the research. Participants then provided verbal informed consent, confirmed by a signed consent form and completed health questionnaire. JHU urine samples were collected from men undergoing prostate biopsy at the Johns Hopkins Hospital with suspicion of prostate cancer. The men were either undergoing their first prostate biopsy or had not been biopsied for more than 1 year prior to collection of the urine sample. Table 1 contains the clinical and pathologic details of the men included in the study. Clean catch urine specimens were obtained and transported for processing within 4 hours of collection. Unprocessed urine was aliquoted into 3 mL aliquots and stored at -80°C until use in this study. Aliquots were used for

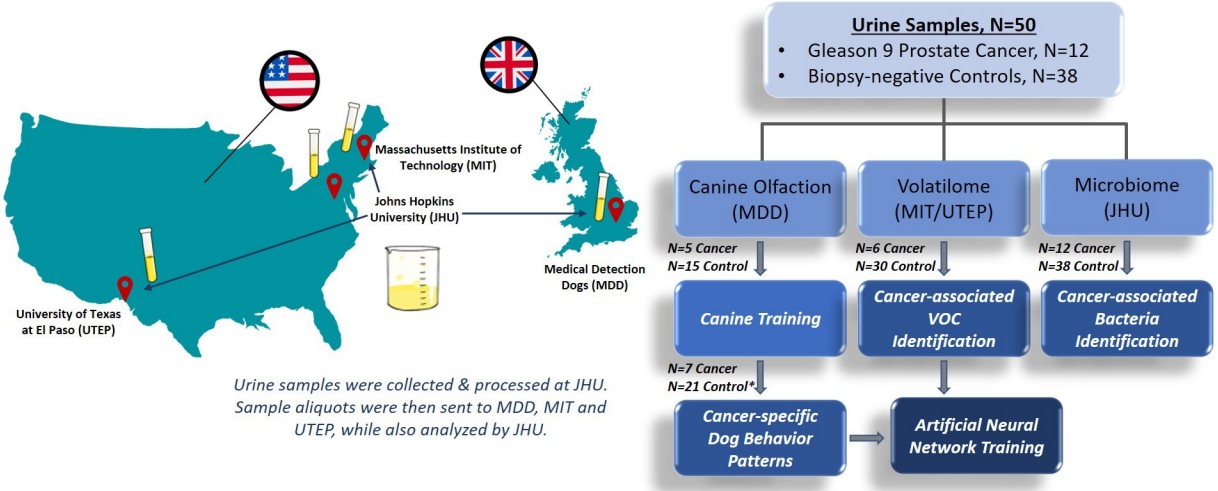

**Fig 1. Study schema of workflow for the analysis of urine samples.** Urine samples from subjects diagnosed with Gleason 9 prostate cancer or biopsy-negative controls were collected and aliquots from each subject were sent for analysis by canine olfaction to Medical Detection Dogs (MDD) in the UK, GC-MS by Massachusetts Institute of Technology (MIT) and University of Texas at El Paso (UTEP) in the US, and microbiota profiling analysis by Johns Hopkins University (JHU) in the US. *Two control samples were reserved as extras for the trial if needed.

canine olfaction studies and for GC-MS. A remaining 30 mL of the urine sample was pelleted by centrifugation at 1000g for 10 minutes and stored at -80˚C. The urine pellets were used in the urine microbiota analyses. The details of the urine samples and use among assays are given in S1 Table.

## Animals

Canine training was performed at Medical Detection Dogs (MDD), UK (https://www.medicaldetectiondogs.org.uk). Two dogs were selected from a pool of six available dogs to participate in this trial after a rigorous selection process. The two dogs selected to participate in the trial were Florin, a 4 year old female Labrador, and Midas, a 7 year old female Wire Haired Hungarian Vizsla (Fig 2A). Each dog had been previously trained on a single source of prostate cancer (positive) samples (Gleason 6–9, stage T1-T4) and control non-cancer (negative) samples from Milton Keynes University Hospital (MKUH) between November 2015 and September 2018. Training using samples supplied by JHU began in October 2018. The details of the canine training results are given in S2 Table.

## Sample storage and preparation for canine olfaction

Frozen urine samples were shipped to MDD from JHU via biosample courier service Biocaire on dry ice, with continuous temperature monitoring indicating no significant temperature fluctuations and upon arrival were expedited through customs in a preserved state. On

**Table 1. Clinical characteristics of the urine samples.**

| Biopsy Diagnosis | Grade | Number of Patients | Median Age in Years (Range, IQR*) | Median PSA in ng/mL (Range, IQR) |
|---|---|---|---|---|
| Biopsy-negative control | | 38 | 58.5 (45–80, 10.5) | 4.7 (1.1–18.4, 4.4) |
| Cancer | Gleason 9 | 12 | 65.5 (49–75, 16.25) | 8.0 (3–76.8, 15.2) |

* Interquartile range.

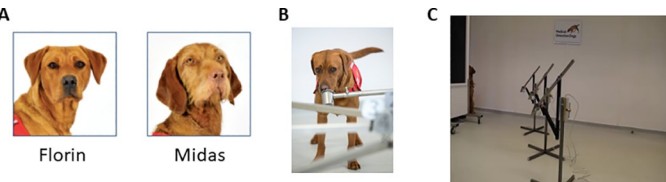

| Dog | Sample Type | Number of Samples | Correct Response | Incorrect Response | Sensitivity (%) | Specificity (%) |
|---|---|---|---|---|---|---|
| Florin | Biopsy-negative control | 21 | 16 | 5 | | 76.2 |
| | Cancer | 7 | 5 | 2 | 71.4 | |
| Midas | Biopsy-negative control | 20 | 14 | 6 | | 70 |
| | Cancer | 7 | 5 | 2 | 71.4 | |

**Fig 2. Study schema for canine olfaction trial.** (A) The two dogs, Florin and Midas, selected to participate in the trial. (B) Image of the presentation pots. (C) Test pots placed into the metal arm attached to the carousel. (D) Comparison of indications to biopsy-negative control and cancer samples in double blind trial. This table shows that out of the 21 control samples, Florin produced 5 false positive indications resulting in 76.2% specificity versus Midas' 6 false positive indications resulting in 70% specificity. Both dogs correctly indicated to 5 out of 7 target samples giving 71.4 sensitivity.

removal from packaging all sample details were cross-checked against the stock list provided and immediately transferred to a -80°C freezer.

One sample at a time was selected and vortexed for ten seconds before being opened. 1 mL of patient urine was decanted into 1.75 mL glass vial. On completion, each glass vial containing a 1 mL aliquot was marked with the anonymized code provided with the sample using a permanent marker. All aliquots of the same code were stored in the same zip-lock bag stored at -80°C. The zip lock bag was also marked with the anonymized code using a permanent marker. All training and testing samples were prepared following this protocol.

Samples were selected and defrosted on the day of training and placed in a refrigerator for no longer that 1 hour before being prepared for the training protocol. An established MDD standard operating procedure to control for cross contamination was followed at all times (See S1 Methods).

## Blinded sample preparation for trial

Urine samples were link anonymized and blinded to MDD with information detailing which samples were to be used for double blind testing. We also received an electronic file from Dr. Steve Morant, Consultant Statistician, University of Dundee and Leslie Mangold MS, Research Administrative Manager, JHU, containing the required samples blinded in set order for exposure to the dogs during the trial. One cancer and three biopsy-negative control samples were included in each run.

On a trial day each sample was selected from the blinded list and prepared for testing following the previously described protocol. The relevant samples were collected from the freezer and defrosted to liquid state. When in liquid state, the pots and aliquot were placed into the refrigerator to stabilize for a minimum of ten minutes before use. When ready to test, each sample was recovered from the refrigerator, decanted into the appropriate glass presentation pot and sealed with a metal lid. This was left for ten minutes before use. For testing, the lid was removed and each test pot was placed into the metal arm attached to the carousel (Fig 2B and 2C) following the blinded predetermined randomization supplied. After presentation to the

dog, the sample was removed from the carousel and resealed with the matched lid until it was required again or was returned to the freezer for future use or storage.

At the beginning of the trial session the DVD recorded CCTV camera was started. The MDD standard operating procedure to control for cross contamination was again followed.

### Canine biodetection performance protocol

At the beginning of every session a number of warm-up runs were completed. These consisted of single blind runs incorporating samples selected from the training cohort to prepare the dogs for work and to satisfy the project specialist that the dogs were ready to complete trial sessions. The trial samples were then prepared according to the sample storage and preparation section above.

Two members of the training team managed the testing: a bio-detection technician, to coordinate the trial, and a project specialist. Also present was the cancer study research leader. Once the test was ready to start, the specialist was called into the room with the dog. The specialist stood behind the shield and tasked the dog to search. The dog approached the carousel at the first position and proceeded anti-clockwise sniffing each position in turn (S1 Movie). If a sample was "indicated" by the dog (e.g., the dog indicated that the sample contained prostate cancer, see S1 Movie), the specialist signaled to the technician coordinating the test, using a hand signal hidden from the dog that this was "called". Florin indicates a sample by standing and staring at the sample whereas Midas sits in front of the sample. The technician consulted the database which revealed the answer, and this was displayed on the dual monitor for the specialist to see. If correct a green tick was displayed and the dog was rewarded appropriately. If incorrect, a red X was displayed, and the dog was recalled from the sample. If a correct response to the test run was "called" the run was deemed finished. The specialist and dog left the room. The samples were recovered, sealed and stored in the refrigerator or freezer according to the requirements of further runs. If the "call" was incorrect, the sample was removed from the line, replaced with a blank and the dog was tasked to search. If a second incorrect "call" was made the run was deemed complete. No further searches took place during this trial with this specific sample set. At any time throughout the testing phase, a calibration run consisting of samples selected and presented from the training cohort could be called by the specialist if it was deemed that the dog had become unsettled.

All runs proceeded in this manner until the trial was completed. When a run had been completed and the result was known, the samples were used by the specialist for calibration of new runs. This enabled us to use relatively novel samples to prepare the dogs for the new test run. In the case of runs where an incorrect decision had been made (false positive more than once), the specialist was unable to use the line to pre-train the dogs as the identity of the positive sample in the run was not known.

### Statistical methods, canine olfaction trial

We calculated the binomial probabilities of the observed success rates in picking positive samples which were presented in sets of four, but each sample required a 'yes/no' decision, based on the null hypothesis that choice was random.

### GC-MS data collection and analysis

GC-MS was coupled with headspace solid-phase microextraction (HS-SPME) to analyze urine samples obtained from prostate cancer patients and biopsy-negative controls. Frozen samples were treated in the same way as described above in the canine section including anonymized coding. Upon thawing to room temperature, samples were transferred via pipette to headspace

vials (Restek, Bellefonte, PA). Volatiles were extracted from the headspace of the urine with carbon wide range SPME arrows (Restek). To facilitate equilibration, the headspace vial-SPME arrow assembly was gently agitated at 172 rpm and 80˚C for 30 minutes.

The SPME arrow fiber was thermally desorbed in the injector of a 6890 GC system coupled with a 5973 mass spectrometer (Agilent Technologies, Palo Alto, CA). The injector was used with a 2:1 split ratio and a 2.0 mL/min split flow at 300˚C. For GC separation, a Zebron$^{TM}$ ZB-624 column (30 m x 0.32 mm x 1.80 mm, 5% cyanopropylphenyl-94% dimethylpolysiloxane, Phenomenex, Torrance, CA) was used, and the carrier gas flow was maintained at 1 mL/min. The oven program was as follows: initial temperature of 40˚C for 1.0 minutes, 10˚C/min ramp up to 300˚C, and 300˚C isothermal for 10 minutes. The MS transfer line temperature was maintained at 240˚C, and MS spectra were recorded in scan mode from m/z (mass to charge ratio) 35–500. GC-MS data was analyzed via Agilent MSD ChemStation software (E.02.02.1431) in combination with MZmine 2 open-source software. The headspace, GC, and MS conditions are summarized in S3 Table.

All detected peaks were screened against the 2017 NIST EPA/NIH mass spectral library using the NIST Mass Spectral Search Program (v2.3, National Institute of Standards and Technology) and the Automated Mass Spectral Deconvolution and Identification System (AMDIS; v2.70, National Institute of Standards and Technology). The mass spectral search produces matches with a match factor that describes the quality of the match. A perfect match results in a value of 999; spectra with no peaks in common result in a value of 0. As a general guide, 900 or greater is an excellent match; 800–900, a good match; 700–800, a fair match. Less than 600 is a poor match. However, unknown spectra with many peaks will tend to yield lower match factors than similar spectra with fewer peaks. Additional identification factors considered include the number of ions in the measured mass spectrum, whether characteristic ions (e.g. molecular ions) are detected, and the number of similar NIST library matches present.

Only the best library match is reported for each peak. This library match should not be considered as definitive identification of an unknown peak. Matches reported here cannot be guaranteed, even when the match quality is high, without additional work including running an identical reference compound under the same conditions.

## Statistical methods, GC-MS

Over 1,157 different VOCs were found in the urine samples, resulting in a high-dimensional modeling problem. To streamline the analysis, we first removed the VOCs that were observed in less than 3% of the entire population. The remaining variables were screened by testing the difference in each VOC between the prostate cancer positive and biopsy-negative control groups. The Wilcoxon rank-sum test was used since it can accommodate the zero inflation among many VOCs. Heat maps were generated to visualize those significant VOCs ($p<0.05$) in the prostate cancer and biopsy-negative control groups.

Applying a liberal cutoff of 0.2 to the p-values, over 29 VOCs remained for the model development. We fit regularized logistic regression models with SIS penalty, and the 10-fold cross-validation was used to select the optimal tuning parameter [24]. The final logistic model was then evaluated via the Receiver Operating Characteristic (ROC) curve and other performance measures on the basis of jackknife prediction [25], which helps alleviate the over-optimism induced by variable selection. Furthermore, Firth's approach was taken to fit the final logistic model in order to achieve bias-reduction for the small sample scenario and deal with the nearly complete separation seen in the data [26]. All statistical analyses are performed using the open-source statistical computing software *R* [27].

## Urinary microbiota profiling and data analysis

DNA was extracted from urine pellet samples (12 Gleason 9 cancer, 38 biopsy-negative controls) and six blank extraction negative controls as previously described [28]. 16S rDNA gene libraries were generated and submitted to the SKCCC Next Generation Sequencing Core at Johns Hopkins for next generation sequencing on an Illumina HiSeq instrument as previously described [28]. Raw paired-end reads were merged into consensus sequences using FLASH requiring a minimum 20 bp overlap and a 5% maximum mismatch density, and subsequently filtered for quality (targeting an error rate < 0.1%) and length (minimum 60 bp) using Trimmomatic and QIIME [29]. Passing sequences were then trimmed of primers, evaluated for chimeras with UCLUST [30] (*de novo* mode), and filtered for host-associated contaminant using Bowtie2 [31] searches of NCBI Homo sapiens Annotation Release 106. Additionally chloroplast and mitochondrial contaminants were detected and filtered using the RDP classifier with a confidence threshold of 50%. High-quality clean 16S sequences (80,000 sequences per sample) were then subjected to high-resolution taxonomic assessment using Resphera Insight [32–34]. One control sample did not have sufficient sequencing reads to continue, therefore the final analysis was performed on 12 cancer samples and 37 biopsy-negative controls.

Contaminant removal was performed in three phases. The first phase identified potential contaminant sequences based on abundance in the negative controls as previously described [28]. The next phase assessed spearman correlation (correlation > 0.30) with 10 indicator contaminant species. Finally, we performed general removal of common contaminant genera (S4 Table). Each sample was normalized via rarefaction to 2,700 clean sequences per sample. Beta-diversity analysis, including Bray-Curtis and UniFrac distance computation and principal coordinates analysis (PCoA), was performed in QIIME. Comparative statistical analysis for differential abundance was performed using the Mann-Whitney U test.

## Neural network training

We trained an ANN to emulate canine cancer diagnoses of urine samples based upon GC-MS data presented to it. We used both network skeletonization [35–37] and auto-associative filtering [38], followed by anomaly detection. Rule extraction techniques were then applied to this net [35–37] to reveal the "logic circuitry" developed within it through both training and the selective pruning of its connection weights. Using this semi-quantitative technique, salient inputs to the model became evident, as well as the interplay between them in determining the ANN prediction of the canine's diagnosis. In general, this approach involves training a network model identifying dominant connection traces bridging outputs of interest with network input nodes they are most functionally dependent upon. Analysis proceeds working backward from the ANN output weight layer, pruning less significant weights to expose the most prominent connective path to the ANN hidden layer and the most important node(s) of that layer. This process is repeated, starting at that node, and working through successive layers of the net until reaching the input layer, the most significant weights therein indicating the critical factors relevant to the chosen output. Driven by the relatively small number of training exemplars available, we trained a preliminary network and skeletonized it in this fashion. The resulting weight pruning indicated the most important GC-MS peaks occurring in the interval from 10 to 14 minutes. Accordingly, we stripped out data outside this range of retention times, thereby reducing the size of the input layer from nearly 9,000 to 205 points. The neural net chosen was a fully connected Multilayer Perceptron (MLP), whose neurons utilized sigmoid transfer functions. This net was trained using a commercial product called PatternMaster$^{TM}$, which features a virtual reality interface that facilitates the rule extraction process, allowing users to "fly" through the net to select network nodes of interest, and then progressively remove connection weights bridging them. Input nodes to this net

represented abundances within the GC-MS data at retention times in the range from 10 to 14 minutes, for a total of 205 data points. The two network output nodes represented canine-indicated positive and canine-indicated negative alerts to cancer. A minimal number of hidden layer nodes were chosen, 32 in all, not to attain optimal prediction accuracy, but to avoid memorization and to facilitate the graphical skeletonization of the net to expose dominant connections.

All GC-MS data, 808 exemplars in all, was normalized to the range [0, 1], using minimum and maximum values of the data. Backpropagation training proceeded with a learning rate was 0.1, and a momentum of 0.03, with random ordinal shuffling of the data occurring with every training epoch to avoid localized learning effects. Scale margin was set to 0.1. The net was trained to a root-mean-square (RMS) error of 0.15, probably the minimal error attainable with this limited set of connection weights.

## Results

### Detection of prostate cancer in urine samples by canine olfaction

Urine samples collected at JHU were sent to MDD, UK (Fig 1). Two dogs, Midas, and Florin, were previously extensively trained to detect prostate cancer and were recalibrated on a test set of urines (5 cancer, 15 biopsy-negative control) from JHU prior to commencement of the pilot trial (see S2 Table and S1 Methods). A double blind pilot trial was then conducted with an additional 7 cancer and 21 biopsy-negative control samples. Fig 2D shows the compiled results of the trial, and Table 2 provides the individual results. Florin correctly identified 16 of the 21 presented biopsy-negative controls as negative for prostate cancer (specificity 76.2%), while Midas correctly identified 14 out of the 20 presented biopsy-negative controls (specificity 70%). Both dogs correctly indicated 5 out of the 7 Gleason 9 prostate cancer samples, resulting in 71.4% sensitivity.

We observed that the training protocol for the dogs had a large effect on their ability to correctly identify samples, consistent with prior research [39]. The two dogs were prepared for double blind testing phase in a two week pre-training period using a forced choice positive bias system where each line contained a positive cancer sample and reinforcement was strongly biased towards indication of a positive cancer sample (e.g., the dogs were only rewarded if they indicated a positive sample). Unlike a traditional forced-choice, a positive bias reward system allows the dogs to leave the search line up if they deemed no positive sample present. The decision to test with the positive bias reward system (i.e. no reward was given for a non-indicated line) was made due to the limited total sample numbers for testing, the sample to control ratio, and the minimal opportunity for the dogs to adapt to samples with new background. However, this decision was reversed partially through testing when it

**Table 2. Individual test results for 7 sample sets each containing one positive cancer and 3 biopsy-negative control samples.**

| | | Florin | | | | Midas | | | | |
|---|---|---|---|---|---|---|---|---|---|---|
| Set | Position | Run 1 | Run 2 | Run 3 | Run 4 | Position | Run 1 | Run 2 | Run 3 | Run 4 |
| 1 | 2 | - | X | ✓ | | 3 | ✓ | | | |
| 2 | 1 | X | X | | | 1 | X | X | | |
| 3 | 3 | - | ✓ | | | 2 | X | ✓ | | |
| 4 | 2 | X | - | - | X | 1 | - | X | X | |
| 5 | 3 | ✓ | | | | 2 | X | - | - | ✓ |
| 6 | 2 | - | ✓ | | | 1 | X | ✓ | | |
| 7 | 3 | - | ✓ | | | 3 | ✓ | | | |

Shaded runs were conducted via a positive bias reward system. Non-shaded runs were conducted via balanced reward system. ✓ = correct indication of cancer sample. X = incorrectly indicated control sample as a cancer sample.–is a run with no indication. Position is the original position of the cancer sample in the set. All outcomes are based on the final pass for each run.

became apparent from the behavior of the dogs that the provided controls were significantly complex. The dogs had very limited prior experience with "biopsy-negative" urine samples from men that likely had other prostatic disease such as BPH and prostatitis. The positive bias reward system was found in fact to be reducing the dog's ability to learn to discriminate. Therefore, the dogs were given a two-week pre-testing preparation period returning to a balanced reward system where true blank lines and finding a target (positive) were rewarded equally. The period that each dog was rewarded in the positive bias reward system is depicted by the shaded area in Table 2.

It is clear from the results, particularly by dog 1 (Florin) that learning had occurred during the first four positive bias reward system sets as on changing to balanced reward in set 5, her results were 100%. The chance of picking the right sample out of 4, three times in a row is $(\frac{1}{4})^3 = 1/64 = 0.016$. The trained canine as a biological detector can adapt rapidly if appropriate reinforcement is given. Dog 2, Midas, also showed improvement, the positive bias reward system period was for only two sets before changing to balanced reward. Although it is not as clear from the data analysis, recorded behavioral changes in the database indicated that her performance was improving as the sets progressed.

Among the eight sample sets presented to either dog using the balanced reward system (Table 2), the cancer sample was incorrectly identified twice in a row by only one of the dogs. The probability of only one, or none, being missed by chance is 0.035.

## Differences in VOC content of cancer versus biopsy-negative control samples

In parallel to the analysis performed by canine olfaction, a subset of the samples (6 cancer, 30 biopsy-negative controls) were sent for VOC analysis by GC-MS (Fig 3A). Only a subset could be used in the final analysis because several of the samples (in a blinded fashion) were used to optimize the GC-MS protocol. By pressurizing and heating the urine samples we ensured that all volatiles available to the dogs' nose at room temperature were also present in the GC-MS experiments. Total raw data is available in the S1 Data. Similarly to previous studies [19, 40], we found individual peaks representing VOCs that were elevated or reduced in prostate cancer versus biopsy-negative control urine samples at p<0.05 (Fig 3B and 3C). VOCs elevated in the Gleason 9 prostate cancer samples were different from those previously reported [19], including trimethyl silanol, a volatile siloxane resulting from the degradation of silicones. However, our study design (Gleason 9 prostate cancer versus biopsy-negative controls) has not been previously examined in regards to urinary VOCs. Several VOCs identified as being decreased in the cancer samples, including 2-pentanone and pyrrole, are commonly found in healthy urine samples [41, 42]. To prevent missing important predictors for prostate cancer prevalence, a relatively large threshold (with p<0.20) was applied to screen variables for further development of the regression model. Over 29 VOCs remained for the development of a logistic model. The final logistic model was evaluated via the Receiver Operating Characteristic (ROC) curve (Fig 3D). On the basis of predicted probabilities from the final model obtained via jack-knife cross-validation, the area under the ROC curve (AUC) is 0.935, indicating a high discriminative power. Further validation of the above regression model using external testing samples is warranted for the development of VOC based diagnostic model.

## Differences in microbial content of cancer versus biopsy-negative control samples

We likewise profiled the urinary microbiota via 16S rDNA sequencing in the samples used for canine olfaction and GC-MS (Fig 1) as previously described [28, 43]. One of the biopsy-negative control samples did not have enough sequencing reads for analysis, therefore the final

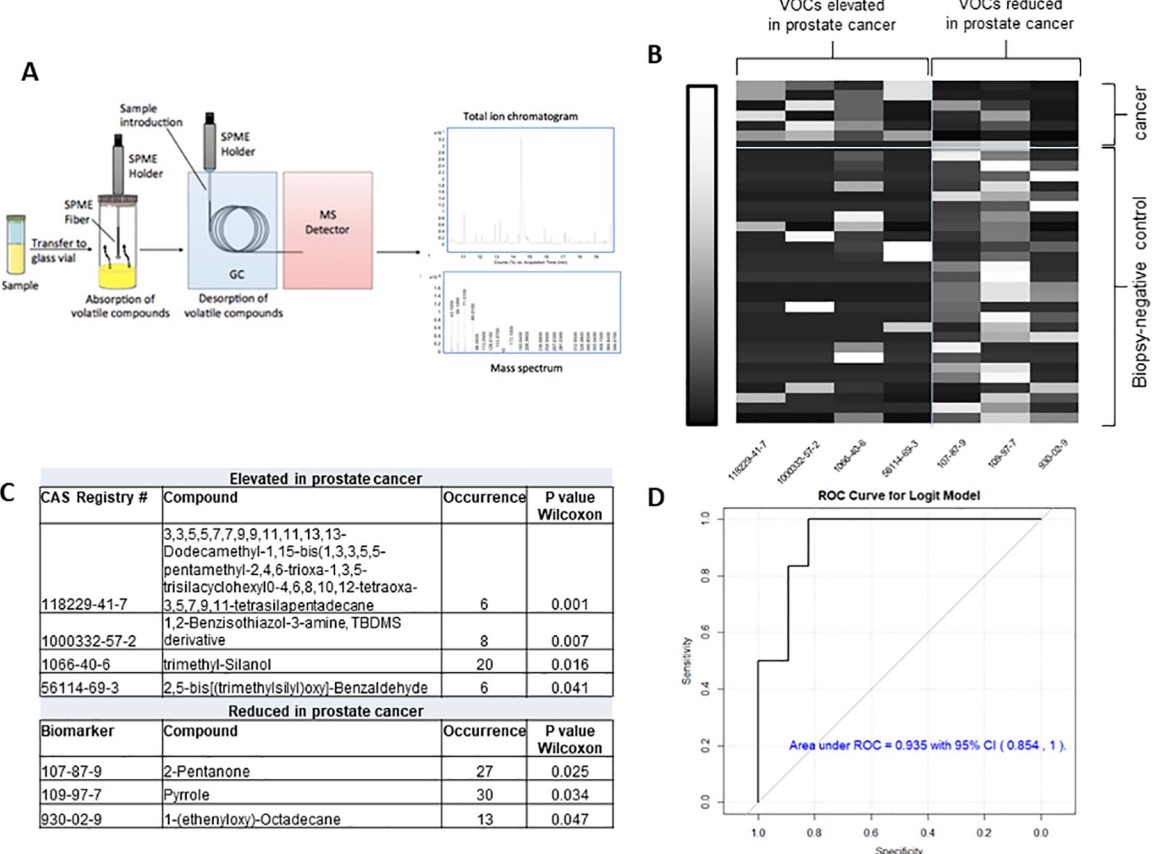

**Fig 3. Analysis of VOCs in patient urine samples.** (A) Study schema for VOC analysis. (B) Heat map of significantly increased or decreased VOCs by Wilcoxon rank-sum test (p <0.05) in cancer versus biopsy-negative control samples. Shown on x-axis are the CAS Registry numbers of the seven significant VOCs (p<0.05) showing elevating or reducing quantity in prostate cancer patients. The correlation between VOCs and patients ranges from low (black) to high (white). (C) Compounds significantly elevated or decreased in cancer versus biopsy-negative control samples. (D) The Receiver Operating Characteristic (ROC) curve for VOC prostate cancer logistic model and verified in 34 patients.

analysis was performed in 12 cancer samples and 37 biopsy-negative control samples. Hierarchical clustering analysis did not indicate a clear separation of cancer versus biopsy-negative control samples based on the complete microbiota profile (Fig 4A). Likewise, beta diversity analyses did not indicate a clear separation of cancer versus biopsy-negative control samples (Fig 4B). Similar to the VOC analyses, there were individual species of bacteria that were differentially abundant in cancer versus biopsy-negative control samples (Fig 4C). One of the bacterial species found to be elevated in Gleason 9 cancer samples, *Dolosigranulum pigrum*, is a rare opportunistic pathogen that has been previously reported in urine samples [44].

Also similar to the VOC analyses, the species of bacteria identified as differentially expressed in cancer versus biopsy-negative control samples were different than what we previously reported [28]. Our previous study included primarily low grade prostate cancer however, and no Gleason 9 prostate cancer urine samples.

## An artificial neural network trained on canine olfaction diagnosis detects differences in cancer versus biopsy-negative controls

On the basis of VOCs collected by headspace SPME and analyzed by GC-MS, the raw ion chromatographs were used to train an ANN to emulate canine cancer diagnoses of urine

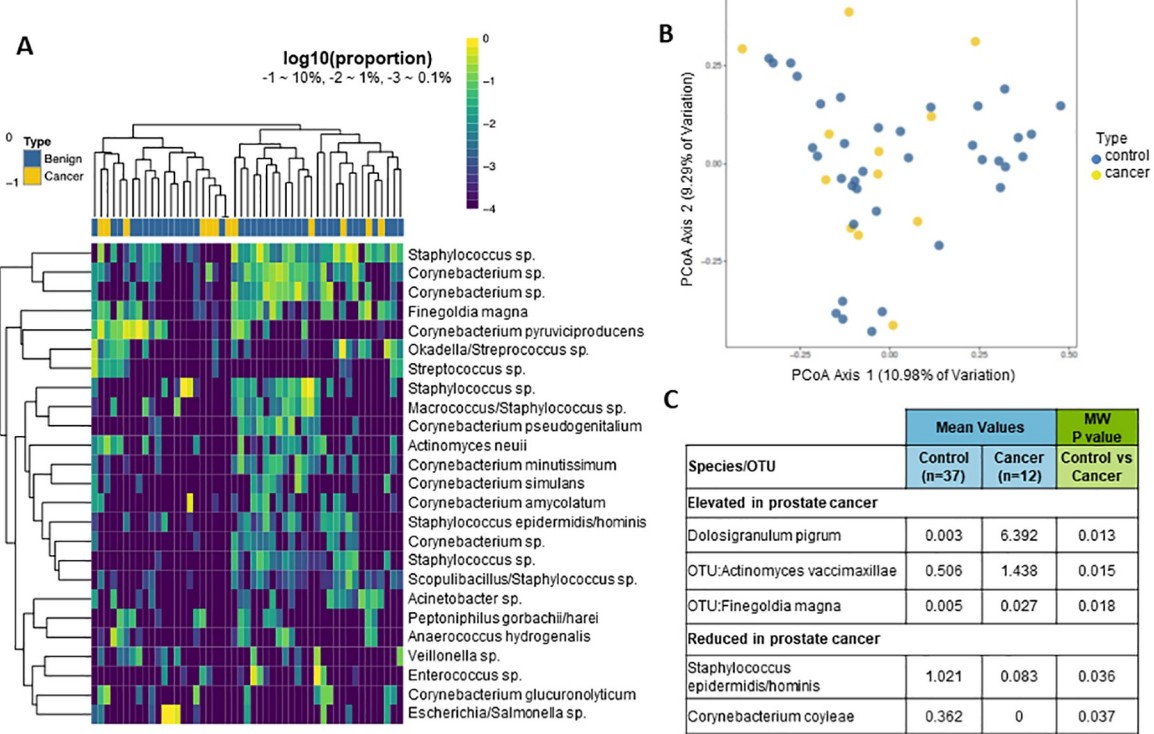

**Fig 4. Analysis of microbiota in patient urine samples.** (A) Unsupervised clustering (log transformed) of 16S rDNA Illumina sequencing results from urine pellet samples by the top 25 species. The dendrogram was based on hierarchical clustering of the Euclidean distance between samples in the combined cancer and biopsy-negative control samples. (B) Beta-diversity (Bray-Curtis) of each urine bacterial profile, analyzed by cancer (yellow) or biopsy-negative control (blue). (C) Differential abundance of select species of bacteria in cancer and biopsy-negative control samples. Mean percent sequence abundances are given for the samples positive for the indicated species from the cancer and biopsy-negative control groups. MW = Mann-Whitney U test.

samples. Both network skeletonization [35–37] and auto-associative filtering [38] techniques were used to reveal the most important chromatograph peaks contributing to the canine diagnosis. The network skeletonization approach (Fig 5) indicated the most salient spectral features occurred in the interval from 10 minutes to 14 minutes. This finding was corroborated by auto-associative filtering (Figs 6 and 7), which indicated the chief differences between cancer and biopsy-negative control urine samples were an abundance of elutes represented by a pair of peaks at 13.177 and 13.563 min, as well as the absence of elutes, relative to biopsy-negative control samples, at 12.698 min. Similar but lesser depletions appear 10.561, 10.899, and 11.473 min. The two techniques (skeletonization and auto-associative filtering) were therefore consistent in indicating the region of the data most important in informing canine diagnosis, and further indicated peaks that were associated with cancer vs. biopsy-negative control urines.

## Discussion

This study demonstrated feasibility and identified the challenges of a multiparametric approach as a first step towards creating a more effective, non-invasive early urine diagnostic method for the highly aggressive histology of prostate cancer (e.g., Gleason grade 9). Canine olfaction was able to discriminate between prostate cancer and biopsy-negative urine samples, and VOC and microbiota profiling analyses showed a qualitative difference between the two groups. Furthermore, an ANN was trained to emulate the canine olfactory diagnoses based on

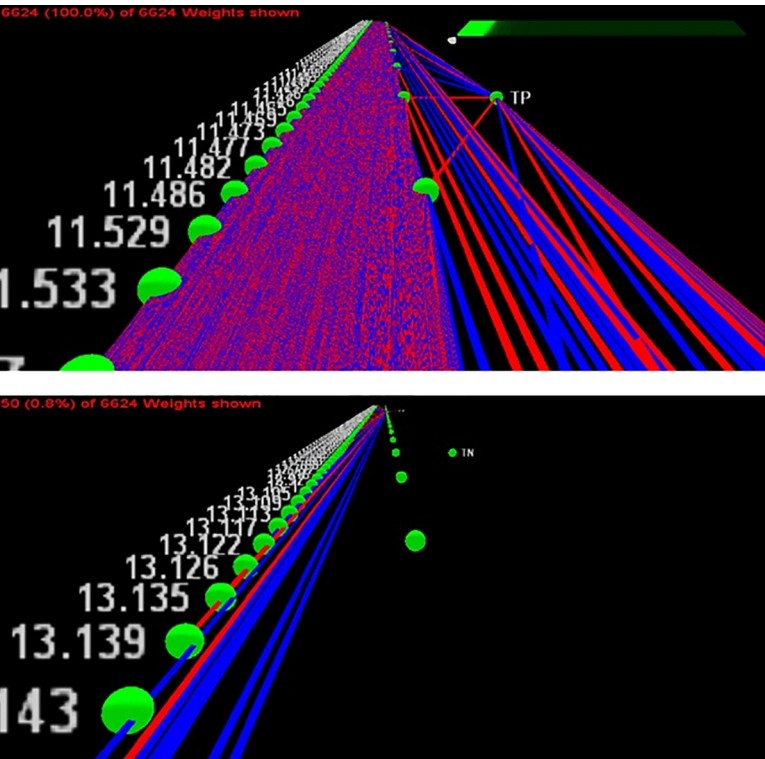

**Fig 5. Network skeletonization of neural net mapping of GC-MS to canine-positive indicated and canine-negative indicated urine samples.** The network is depicted as a system of excitatory (red) and inhibitory connection weights (blue). Starting from the output node representing a canine-indicated positive (TP) canine diagnosis of prostate cancer, less significant weights are stripped away to reveal critical connections to the most dominant GC-MS peaks contributing to the canine cancer diagnosis. The top figure shows the net with all weights present, while the bottom figure reveals the peak near 13.139 minutes as positively correlated (i.e., red connection) with canine-positive indication of prostate cancer.

GC-MS analytical data. Our results indicate that there may be information synthesized by the dogs regarding the nature of cancer that may not be readily identified by traditional single channel molecular biomarker analysis, and may instead be an emergent property [21].

Although tested on a small sample set which does not enable us to make definitive conclusions about accuracy, the results achieved in this pilot support the potential of specialist trained detection dogs directly assisting in the development of an ANN to run on a bio-electronic machine olfaction diagnostic device [23]. Our results demonstrate the canine ability to discriminate, learn, and improve detection even when presented a small number of samples of a complex odor. The challenge remains on how to port canine intelligence into machine olfactors [45].

Dogs are exceptional at scent discrimination, and are also known for their ability to recognize tiny changes in odor background. Changes in odor background often result in a detectable change of behavior, before habituation occurs. It is important therefore to habituate to odors that may occur in samples held in a different facility produced in a different part of the world, before being challenged with difficult discrimination decisions. Our pilot study was limited by the number of urine samples available at JHU from men with Gleason 9 prostate cancer with enough aliquots as well as a urine pellet available to perform our three-armed study. The limited sample size was this study's biggest overall challenge and particularly significant here, as we were asking dogs to discriminate more complex samples than they had ever

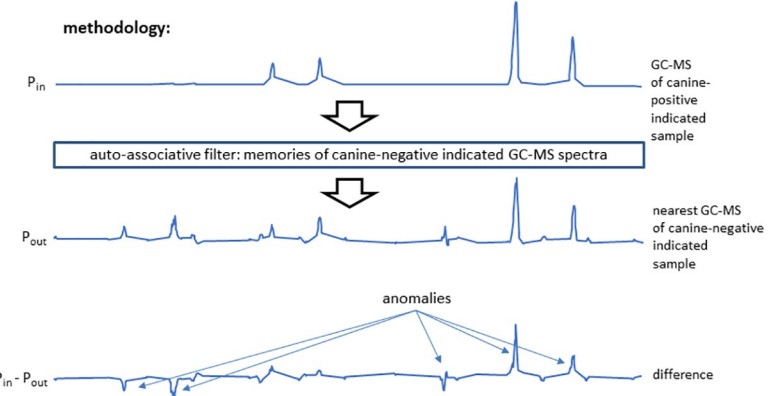

**Fig 6. Auto-associative filtering methodology.** An auto-associative net was trained to reconstruct the GC-MS spectra of all the canine-negative indicated samples. Inputting the spectra of canine-positive indicated spectra, the net generated the nearest canine-negative indicated spectrum at its output. Subtraction of the output from the input spectrum revealed anomalous features possibly associated with the canine indication of cancer. In the example shown, both elute excesses (peaks) and deficiencies (troughs) are indicated in the difference spectrum. In short, this network acts as a database lookup table that supplies the closest matching canine-negative indicated spectrum to one that is applied, if need be producing synthetic data representing a potential canine-negative spectrum.

previously received training on. Following detailed discussion, we decided to use five cancer samples for familiarization training, using the remaining seven for testing.

Interestingly, two biopsy-negative controls (JHBUI-2719 and AWP-9582) were picked incorrectly by both dogs independently in testing, and in the same order despite double blind and random position (Table 2). GC-MS was not performed on these samples, however the microbiota sequencing yielded interesting results. JHBUI-2719 had an abundance of *Alloscardovia omnicolens* (21.5% of sequencing reads) which wasn't present in any of the other cancer samples. AWP-9582 had an atypical urinary microbiome profile where almost all of the reads were assigned to either *Dolosigranulum pigrum* (found to be more abundant in cancer samples) or an unknown species of *Lactobacillales*. Understanding the reason for the dogs' error, and whether urinary microbiota contributed to the error, could provide valuable insights and it is our plan to investigate it in future, larger sample size studies.

In this pilot study, the dogs were also being tested against particularly difficult controls: urine from men that were biopsy-negative for prostate cancer. These men would all have been biopsied for an indication of prostate cancer (most typically elevated PSA or abnormal DRE). Therefore, the control group likely had other prostatic disease such as BPH and prostatitis, and likewise it is possible that some of the biopsy-negative men actually had prostate cancer that was missed on biopsy. Unfortunately, follow up data on any subsequent positive biopsies for these men was not available to us. The dogs had only limited training to discriminate biopsy-negative controls from prostate cancer in the past due to the difficulty of sourcing well annotated samples of this from our MKUH collaborator. We have since determined that it is essential not to attempt to train discrimination unless it can be confirmed with certainty that the control is cancer-free. We therefore submit that testing dogs to this level of performance warrants a significant amount of training against this control group, and this must be a focal point of all future larger-scale studies.

Initially, we made the decision to use 'positive bias reward system' response from the dog. This requires the dog to continue around the carousel until he or she has made a detection decision. The two dogs used were not originally trained under this protocol but could still signal 'blank' (i.e. no decision made). We shifted positive bias to expectation of a prostate cancer in every line. Unfortunately, this also increased the likelihood of false positives. After the

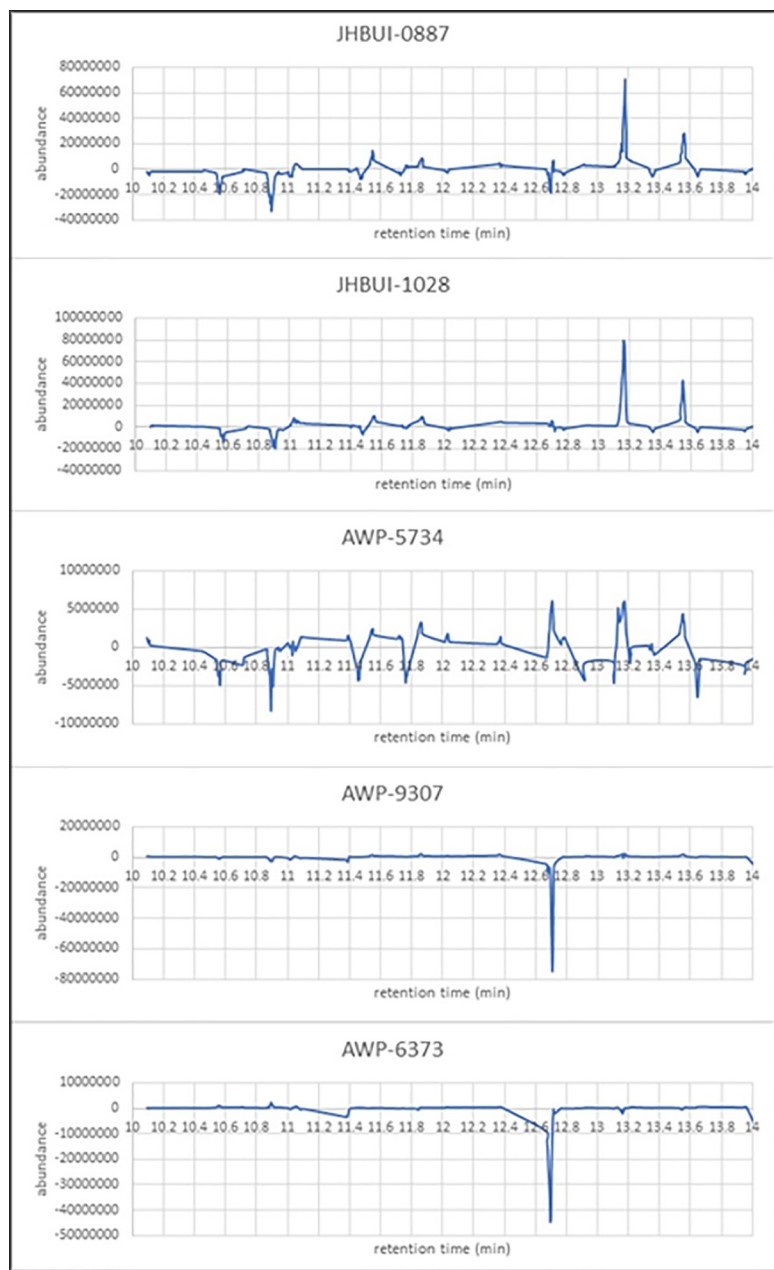

**Fig 7. Auto-associative filtering reveals the most dominant GC-MS features contributing to canine-positive indication of prostate cancer from urine samples.** Anomalies fall into two groups: those showing an overabundance of elutes (JHBUI-0887, JHBUI-1028, and AWP-5734) and those revealing depletions of elutes (AWP-9307 and AWP-6373). The peak near 13.2 minutes in the first three of these plots corresponds to that resolved at 13.139 minutes via network skeletonization.

exposure to the first set of testing, we re-visited this decision and agreed this had been an incorrect choice. We therefore recalibrated the dogs to accept all negative lines, hence lowering false positive bias.

In addition to the above, ours and independent group's published research indicates that dogs do require further training when novel background odors are introduced, through a

change of sample processing or environmental background. This training is necessary to ensure the dogs are confident that these changing background odors are indeed irrelevant and are to be ignored. This method does not require many presentations and it is an important adjustment to enable the dog to re-habituate prior to complex discrimination tasks.

We performed chemical analyses of VOCs, which were able to identify VOC species differentiating cancer from biopsy-negative control samples, however these molecular species were distinct from those previously reported [40, 46]. The very nature of the GC-MS technique can be seen as being loosely analogous to biological olfaction: GC-MS peaks are not always correspondent to just one molecular species and fragments from different molecules but of equal charge/mass ratios might add to the same peak while fragments of the same molecule might appear as parts of various peaks. Similarly, most canine olfactory receptors are tuned to respond to more than one volatile and the same volatile typically activates more than one receptor to various levels with different receptor binding often dependent on different parts of an odorant. In GC-MS the pattern of peaks is used to identify compounds, the "signature" of identity is inferred, and the context of what other molecules are present or absent can greatly affect how the identity of that molecule is interpreted. Similarly, in biological olfaction the presence of some odorants or even odorless volatiles and combinations of odorants in mixtures can affect the scent character detected [22, 47] by directly or indirectly affecting the EC50 activation levels of individual receptors acting as allosteric or orthosteric modulators. The way a scent signature of cancer or anything else a dog is trained to look for is encoded in the brain [21], not by a particular set of molecules identified by name and concentration, but a gestalt representation of many observations in a flexible context. Thus, in contrast to earlier studies of the urinary volatilome in prostate cancer, our primary goal was not to identify compounds differing between prostate cancer and biopsy-negative control urines. Rather, we focused on identifying common emergent properties signaling cancer versus non-cancer that could then be ported to a machine learning platform, giving potential for diagnoses that do not depend on individual biomarkers.

Much work has been done in the past by groups including our own to determine a specific set of VOC signature biomarkers (by name and potentially relative or absolute concentration). Efforts to mine the volatilome for robust molecular species signatures indicating a number of conditions including prostate [46], colorectal [48], liver [49], and lung cancers [48] have consistently yielded schemes where a set of selection rules can be determined cleaving any sufficiently rich data set into positive and negative control clusters. The emergent issue is that just as seen with canines, it appears that no unique (set of) molecule(s) can be found as unambiguous biomarkers and instead it is the scent character, as opposed to the molecular composition of scent that communicates the cancer signature to canines. GC-MS produces a list of VOCs that carry the information for each experimental case but such lists themselves unfortunately do not generalize to provide a "master biomarker template" in the form of molecular species identifications and concentrations [48]. This shows the potential for prostate cancer diagnoses, but validation of the VOCs remains as future research objectives.

We further report microbiota analysis of cancer versus healthy urines which did not identify a clear separation between cancer and healthy urine, although there were individual species that distinguished the two groups. In the microbiome, similarly, there may be particular roles in a microbial community that can be filled by different organisms; it may be the role that is altered in cancer, and determining the characteristics of the role may be more important than determining the organisms.

The purpose of this study was to demonstrate the feasibility of integrating the analysis of prostate cancer patient urine samples using canine olfaction, with chemical and microbial profiling. Diagnostic tests beyond PSA have been established in the last few years and are relevant

to the planned second phase of this study which will aim to compare the receiver operating curves of integrating canine olfaction and chemical and microbial profiling with diagnostic tests such as Prostate Health Index (phi), SelectMDx, and PCA3. As in the case of PSA, these are specific molecular ID biomarker-driven diagnostic metrics while the olfactory approach central to our study depends on the emergent property of scent character. The olfactory diagnostic signal is not necessarily bound to an immutable (set of) molecular identities(s) as the same functional scent character can be conferred to the dogs and machine olfactors via a variety of VOC cocktails. We do note that biomarker-driven as well as emergent-property powered diagnostic metrics, examined collectively, are reasonably expected to paint a more accurate picture of the condition and is likely that their usefulness in directing intervention decisions can be enhanced with the addition of a microbial profiling diagnostic examined here. This integrative approach is a focal point for our future research direction.

Finally, we examined the feasibility, challenges, and opportunities of using canine olfaction diagnosis to train an ANN to characterize its performance in distinguishing between cancerous and non-cancerous (i.e. biopsy-negative control) states using GC-MS data alone as well as in combination with the canine data collected on same samples.

In conclusion, our data speaks to the feasibility of discriminating Gleason 9 prostate cancer from biopsy-negative controls by integrative analysis of several vastly different methodologies, each of which has been shown capable to various degrees by itself: trained canine olfaction, conventional GC-MS analysis of urine headspace VOC as well as our novel, purpose-developed ANN approach, and urinary microbiota profiling on the same samples. The canines were able to detect Gleason 9 prostate cancer versus biopsy-negative controls at a high sensitivity and specificity. Analysis of GC-MS data collected on urinary VOCs was able to identify molecular species differentiating cancer from biopsy-negative controls while further validation is needed. Microbiota profiling did not differentiate prostate cancer from biopsy-negative controls when assessed as a whole, however individual VOCs and microbial species were found to be differentially abundant in the two groups. Combining these data streams allowed us to train an ANN to emulate canine olfactory diagnosis. The biggest challenge throughout this pilot study was the availability of pathologically well-characterized and standardized urine samples. We were aware of this limitation and designed the study accordingly fully expecting that the small number of samples will prevent us achieving the very high sensitivity and selectivity that has been shown to be generally achievable with canines, and similarly limited our training of the ANN. We fully expect that larger sample pools will be the key enabler of statistically powered, multi-institutional future studies seeking to fully integrate VOC and microbiota profiling. The end goal of the pilot study we report here has been to pave the way towards development of machine-based olfactory diagnostic tools that define and recapitulate what can be detected and accomplished now via canine olfaction.

## Supporting information

**S1 Table. Sample use among canine olfaction, GC-MS/ANN, and microbial profiling.**
(DOCX)

**S2 Table. Summary of canine training data over 22 training days.** Number and percent correct after each pass.
(DOCX)

**S3 Table. Instrument parameters for HS GC-MS testing of urine samples.**
(DOCX)

**S4 Table. Common contaminant genera in microbiome analyses.**
(DOCX)

**S1 Methods.**
(DOCX)

**S1 Movie.**
(MP4)

**S1 Data.**
(CSV)

**S2 Data.**
(CSV)

**S3 Data.**
(CSV)

**S4 Data.**
(CSV)

**S5 Data.**
(CSV)

**S6 Data.**
(CSV)

**S7 Data.**
(CSV)

**S8 Data.**
(CSV)

**S9 Data.**
(CSV)

**S10 Data.**
(CSV)

**S11 Data.**
(CSV)

**S12 Data.**
(CSV)

**S13 Data.**
(CSV)

**S14 Data.**
(CSV)

**S15 Data.**
(CSV)

**S16 Data.**
(CSV)

**S17 Data.**
(CSV)

**S18 Data.**
(CSV)

**S19 Data.**
(CSV)

**S20 Data.**
(CSV)

**S21 Data.**
(CSV)

**S22 Data.**
(CSV)

**S23 Data.**
(CSV)

**S24 Data.**
(CSV)

**S25 Data.**
(CSV)

**S26 Data.**
(CSV)

**S27 Data.**
(CSV)

**S28 Data.**
(CSV)

**S29 Data.**
(CSV)

**S30 Data.**
(CSV)

**S31 Data.**
(CSV)

**S32 Data.**
(CSV)

**S33 Data.**
(CSV)

**S34 Data.**
(CSV)

**S35 Data.**
(CSV)

**S36 Data.**
(CSV)

## Acknowledgments

We thank Prof. Neil Gershenfeld of the Center for Bits and Atoms (CBA) at MIT, are indebted to Joe Murphy, Kara Pendlebury and James Prue for administrative support. Dr. Adam Feldman, Department of Urology at Massachusetts General Hospital hosted group discussions and led lab tour on clinical urology practices, including urine sample collection and characterization. Prof. Federico Casalegno of MIT's Design Lab and Samsung for discussions on emerging smartphone noses. We thank Dr. Hanno Steen at Boston Children's Hospital for sharing knowledge on urine biomarkers. We thank Dr. Marvin Weinstein and Dr. Bernard Chen of Quantum Insights for exploring a particularly promising new clustering method called "Dynamic Quantum Clustering". Prof. Ann-Sophie Barwich of Indiana University for seminal exchange of ideas on recognition of olfactory objects. Dr. Rich Fletcher and Carolyn Jin of MIT Media Lab and D-Lab for though-provoking musification of data and Alex Spiliotopoulos of Orion Data Sciences for useful discussions on autostereogram presentation of olfactory and GC-MS data. We thank and acknowledge Dr. James White of Resphera Biosciences for assistance with microbiome sequencing analysis. We are grateful to The Bodossakis' Foundation support of P.M. S. We thank Alfonso Parra Rubio for assisting with Fig 1.

## Author Contributions

**Conceptualization:** Claire Guest, Rob Harris, Karen S. Sfanos, Bruce Trock, Jonathan Simons, Howard Soule, Thomas Johnson, Wen-Yee Lee, Stephen Thaler, Andreas Mershin.

**Formal analysis:** Karen S. Sfanos, Bruce Trock, Rebecca Bader, Adam Kozak, Wen-Yee Lee, Stephen Thaler, Andreas Mershin.

**Funding acquisition:** Karen S. Sfanos, Jonathan Simons, Howard Soule, Thomas Johnson, Wen-Yee Lee, Andreas Mershin.

**Investigation:** Claire Guest, Rob Harris, Karen S. Sfanos, Eva Shrestha, Rebecca Bader, Adam Kozak, Scott Mclean, Wen-Yee Lee, Qin Gao, Sophie Aziz, Patritsia Maria Stathatou, Stephen Thaler, Andreas Mershin.

**Methodology:** Claire Guest, Rob Harris, Karen S. Sfanos, Eva Shrestha, Bruce Trock, Rebecca Bader, Adam Kozak, Scott Mclean, Jonathan Simons, Howard Soule, Wen-Yee Lee, Patritsia Maria Stathatou, Stephen Thaler, Andreas Mershin.

**Project administration:** Claire Guest, Rob Harris, Karen S. Sfanos, Rebecca Bader, Adam Kozak, Jonathan Simons, Thomas Johnson, Wen-Yee Lee, Andreas Mershin.

**Resources:** Alan W. Partin, Leslie Mangold.

**Supervision:** Claire Guest, Karen S. Sfanos, Alan W. Partin, Jonathan Simons, Howard Soule.

**Writing – original draft:** Claire Guest, Rob Harris, Karen S. Sfanos, Bruce Trock, Jonathan Simons, Thomas Johnson, Wen-Yee Lee, Qin Gao, Sophie Aziz, Patritsia Maria Stathatou, Stephen Thaler, Simmie Foster, Andreas Mershin.

**Writing – review & editing:** Claire Guest, Rob Harris, Karen S. Sfanos, Eva Shrestha, Alan W. Partin, Bruce Trock, Leslie Mangold, Rebecca Bader, Adam Kozak, Scott Mclean, Jonathan Simons, Howard Soule, Thomas Johnson, Wen-Yee Lee, Qin Gao, Sophie Aziz, Patritsia Maria Stathatou, Stephen Thaler, Simmie Foster, Andreas Mershin.

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
