## [Decision Letter · Decision Letter 0]

23 Sep 2020

PONE-D-20-26070

Feasibility of Integrating Canine Olfaction with Chemical and Microbial Profiling of Urine to Detect Lethal Prostate Cancer

PLOS ONE

Dear Dr. Mershin,

Thank you for submitting your manuscript to PLOS ONE. After careful consideration, we feel that it has merit but does not fully meet PLOS ONE’s publication criteria as it currently stands. Therefore, we invite you to submit a revised version of the manuscript that addresses the points raised during the review process.

The scientific presentation of the manuscript is according to the Journal´s criteria and only minor revisions as indicated by the Reviewer are requested.

We look forward to receiving your revised manuscript.

Kind regards,

Zoran Culig

Academic Editor

PLOS ONE

Journal Requirements:

We note that one or more of the authors are employed by a commercial company: Imagination Engines.

3.1. Please provide an amended Funding Statement declaring this commercial affiliation, as well as a statement regarding the Role of Funders in your study. If the funding organization did not play a role in the study design, data collection and analysis, decision to publish, or preparation of the manuscript and only provided financial support in the form of authors' salaries and/or research materials, please review your statements relating to the author contributions, and ensure you have specifically and accurately indicated the role(s) that these authors had in your study. You can update author roles in the Author Contributions section of the online submission form.

3.2. Please also provide an updated Competing Interests Statement declaring this commercial affiliation along with any other relevant declarations relating to employment, consultancy, patents, products in development, or marketed products, etc.  

5. We note that Figure 1 in your submission contain map images which may be copyrighted. All PLOS content is published under the Creative Commons Attribution License (CC BY 4.0), which means that the manuscript, images, and Supporting Information files will be freely available online, and any third party is permitted to access, download, copy, distribute, and use these materials in any way, even commercially, with proper attribution. For these reasons, we cannot publish previously copyrighted maps or satellite images created using proprietary data, such as Google software (Google Maps, Street View, and Earth). For more information, see our copyright guidelines: http://journals.plos.org/plosone/s/licenses-and-copyright.

5.1.    You may seek permission from the original copyright holder of Figure 1 to publish the content specifically under the CC BY 4.0 license. 

5.2.    If you are unable to obtain permission from the original copyright holder to publish these figures under the CC BY 4.0 license or if the copyright holder’s requirements are incompatible with the CC BY 4.0 license, please either i) remove the figure or ii) supply a replacement figure that complies with the CC BY 4.0 license. Please check copyright information on all replacement figures and update the figure caption with source information. If applicable, please specify in the figure caption text when a figure is similar but not identical to the original image and is therefore for illustrative purposes only.

Reviewers' comments:

Reviewer's Responses to Questions

**Comments to the Author**

1. Is the manuscript technically sound, and do the data support the conclusions?

Reviewer #1: Yes

2. Has the statistical analysis been performed appropriately and rigorously? 

Reviewer #1: Yes

3. Have the authors made all data underlying the findings in their manuscript fully available?

Reviewer #1: Yes

4. Is the manuscript presented in an intelligible fashion and written in standard English?

Reviewer #1: Yes

5. Review Comments to the Author

Reviewer #1: In the present pilot study „Feasibility of Integrating Canine Olfaction with Chemical and Microbial Profiling of Urine to Detect Lethal Prostate Cancer” the authors describe the possibility to detect high risk prostate cancer by canine olfaction. In addition, the authors tried to identify volatile organic compounds (VOC) and a microbial profile responsible for the cancer specific olfaction. To this end 16S rDNA sequenzing and gas chromatography-mass spectroscopy (GC-MS) artificial neural network (ANN)-assisted examination have been performed.

The data is presented in a way that is clearly understandable and the manuscript is well written, the references are current and accurate. The methods confirms that there is potential in the creation of a VOC profile, as it has been suggested in different cancer entities. This verification was the declared aim of the authors, and therefore further experimental approaches are obsolete. Limitation of the study is the small sample size, however, the authors indicate that the presented study was a pilot study and a study with increased sample size will follow.

However, since there should be some minor changes in the publication, I suggest the acceptance of the manuscript after minor revision. The individual points are listed below:

- Several markers and tests have been established (e.g. Prostate Health Index (phi), SelectMDx, PCA3) the last years. The authors should discuss and compare their findings with other biomarkers and tests.

- The authors should discuss if their “VOC signature” presented in this study correlates with signatures identified in other cancers to distinguish between a PCa signature and a general cancer signature.

6. PLOS authors have the option to publish the peer review history of their article (what does this mean?). If published, this will include your full peer review and any attached files.

Reviewer #1: No

---

## [Author Response · Author response to Decision Letter 0]

14 Dec 2020

Response to Reviewers/Rebuttal Letter for: “Feasibility of Integrating Canine Olfaction with Chemical and Microbial Profiling of Urine to Detect Lethal Prostate Cancer” 

Dear Dr. Culig, 

We are thrilled to receive the positive feedback and thoughtful comments of the reviewers on our manuscript. We are indebted to their and your efforts and, as requested have carefully addressed each point as summarized below and have uploaded the marked-up and unmarked versions of the edited manuscript.

1. We have ensured the manuscript meets PLOS ONE’s style requirements including for file naming.

2. Additional details regarding participant consent have been added to the methods section. Specifically, we have clarified that: “US-originated urine samples used in the final canine training and all GC-MS and canine testing, were obtained under a Johns Hopkins University (JHU) Medicine Institutional Review Board (IRB) approved protocol with written informed consent. For early training and canine testing the UK-originated samples were collected from participants identified by a member of the clinical team who provided them a verbal explanation and information sheet outlining the research. Participants then provided verbal informed consent, confirmed by a signed consent form and completed health questionnaire.” 

3. We confirm the following proposed statements regarding the funding statement and competing interest statement are accurate and suitable to appear alongside the manuscript.

3.1&3.2 We confirm the following proposed statement is accurate and suitable to appear alongside the manuscript. We have updated the author roles in the Author Contributions section of the online submission form and have included an amended Funding Statement as follows: "This work was funded by the Prostate Cancer Foundation Grant (18PILO02) received by CG, AM, and KS. PCF provided partial salary support for authors CG, RH, PMS, AM, and had a role in the study design and preparation of the manuscript, but had no role in the data collection and analysis or the decision to publish. The National Cancer Institute of the National Institutes of Health provided support for WYL and QG (Award Number SC1CA245675). Imagination Engines, Inc. provided support in the form of salary for ST. The NCI NIH and Imagination Engines, Inc. played a role in study design, analysis, decision to publish, and preparation of the manuscript. The specific roles of these authors are articulated in the ‘author contributions’ section."

We confirm the following proposed statement is accurate and suitable to appear alongside the manuscript: "The authors have read the journal’s policy and have the following competing interests: ST is a paid employee of Imagination Engines, Inc. This does not alter our adherence to PLOS ONE policies on sharing data and materials. There are no patents, products in development or marketed products associated with this research to declare.”

4. We have added the corresponding author’s (AM) ORCID# [0000 0001 8194 6241] 

5. The map as well as all other graphics in Figure 1 (i.e. the USA & UK maps and flags, beaker and test tubes) were created by MIT Center for Bits and Atoms student, Alfonso Para Rubio. An acknowledgment of thanks for Alfonso has been added to the manuscript.

Reviewer #1’s comment 1: 

“ Several markers and tests have been established (e.g. Prostate Health Index (phi), SelectMDx, PCA3) the last years. The authors should discuss and compare their findings with other biomarkers and tests.”

In response to this comment we have now added the required discussion to the text: 

The purpose of this study was to demonstrate the feasibility of integrating the analysis of prostate cancer patient urine samples using canine olfaction, with chemical and microbial profiling. Diagnostic tests beyond PSA have been established in the last few years and are relevant to the planned second phase of this study which will aim to compare the receiver operating curves of integrating canine olfaction and chemical and microbial profiling with diagnostic tests such as Prostate Health Index (phi), SelectMDx, and PCA3. As in the case of PSA, these are specific molecular ID biomarker-driven diagnostic metrics while the olfactory approach central to our study depends on the emergent property of scent character. The olfactory diagnostic signal is not necessarily bound to an immutable (set of) molecular identities(s) as the same functional scent character can be conferred to the dogs and machine olfactors via a variety of VOC cocktails. We do note that biomarker-driven as well as emergent-property powered diagnostic metrics, examined collectively, are reasonably expected to paint a more accurate picture of the condition and is likely that their usefulness in directing intervention decisions can be enhanced with the addition of a microbial profiling diagnostic examined here. This integrative approach is a focal point for our future research direction. 

Reviewer #1’s comment 2: 

“The authors should discuss if their “VOC signature” presented in this study correlates with signatures identified in other cancers to distinguish between a PCa signature and a general cancer signature”

In response to reviewer’s comment 2 we have added the following to the discussion:

Much work has been done in the past by groups including our own to determine a specific set of VOC signature biomarkers (by name and potentially relative or absolute concentration). Efforts to mine the volatilome for robust molecular species signatures indicating a number of conditions including prostate [46], colorectal [48], liver [49], and lung cancers [48] have consistently yielded schemes where a set of selection rules can be determined cleaving any sufficiently rich data set into positive and negative control clusters. The emergent issue is that just as seen with canines, it appears that no unique (set of) molecule(s) can be found as unambiguous biomarkers and instead it is the scent character, as opposed to the molecular composition of scent that communicates the cancer signature to canines. GC-MS produces a list of VOCs that carry the information for each experimental case but such lists themselves unfortunately do not generalize to provide a “master biomarker template” in the form of molecular species identifications and concentrations [48]. This shows the potential for prostate cancer diagnoses, but validation of the VOCs remains as future research objectives. 

Sincerely, on behalf the co-authors

Andreas Mershin, Ph.D.

---

## [Editor Report · Decision Letter 1]

4 Jan 2021

Feasibility of Integrating Canine Olfaction with Chemical and Microbial Profiling of Urine to Detect Lethal Prostate Cancer

PONE-D-20-26070R1

Dear Dr. Mershin,

We’re pleased to inform you that your manuscript has been judged scientifically suitable for publication and will be formally accepted for publication once it meets all outstanding technical requirements.

Kind regards,

Zoran Culig

Academic Editor

PLOS ONE
---

## [Editor Report · Acceptance letter]

18 Jan 2021

PONE-D-20-26070R1 

Feasibility of Integrating Canine Olfaction with Chemical and Microbial Profiling of Urine to Detect Lethal Prostate Cancer 

Dear Dr. Mershin:

I'm pleased to inform you that your manuscript has been deemed suitable for publication in PLOS ONE. Congratulations! Your manuscript is now with our production department. 

Kind regards, 

on behalf of

Dr. Zoran Culig 

Academic Editor

PLOS ONE